# Regulation of Protein-Induced Apoptosis and Autophagy in Human Hepatocytes Treated with Metformin and Paclitaxel In Silico and In Vitro

**DOI:** 10.3390/biomedicines11102688

**Published:** 2023-09-30

**Authors:** Norah Saeed Al-Zahrani, Mazin Abdulaziz Zamzami, Mohammed A. Baghdadi, Afnan H. El-Gowily, Ehab M. M. Ali

**Affiliations:** 1Department of Clinical Biochemistry, Collage of Medicine, King Khalid University, Abha 61421, Saudi Arabia; nalzahrani@kku.edu.sa; 2Department of Biochemistry, Faculty of Science, King Abdulaziz University, Jeddah 21589, Saudi Arabia; 3Research Center, King Faisal Specialist Hospital and Research Center, Jeddah 21499, Saudi Arabia; mbaghdadi@kfshrc.edu.sa; 4Research Group “Cancer, Haemostasis and Angiogenesis”, INSERM U938, Saint-Antoine Research Center, University Institute of Cancerology, Faculty of Medicine, Sorbonne University, 75012 Paris, France; 5Division of Biochemistry, Chemistry Department, Faculty of Science Tanta University, Tanta 31527, Egypt; afnan.hamdy@science.tanta.edu.eg

**Keywords:** metformin, paclitaxel, AMPK-α, apoptosis, autophagy

## Abstract

Metformin and paclitaxel therapy offer promising outcomes in the treatment of liver cancer. Combining paclitaxel with metformin enhances treatment effectiveness and mitigates the adverse effects associated with paclitaxel alone. This study explored the anticancer properties of metformin and paclitaxel in HepG2 liver cancer cells, MCF-7 breast cancer cells, and HCT116 colon cancer cells. The results demonstrated that the combination of these agents exhibited a lower IC50 in the tested cell lines compared to paclitaxel monotherapy. Notably, treating the HepG2 cell line with this combination led to a reduction in the G0/G1 phase and an increase in the S and G2/M phases, ultimately triggering early apoptosis. To further investigate the interaction between the cellular proteins with paclitaxel and metformin, an in silico study was conducted using proteins chosen from a protein data bank (PDB). Among the proteins studied, AMPK-α, EGFRK, and FKBP12-mTOR exhibited the highest binding free energy, with values of −11.01, −10.59, and −15.63 kcal/mol, respectively, indicating strong inhibitory or enhancing effects on these proteins. When HepG2 cells were exposed to both paclitaxel and metformin, there was an upregulation in the gene expression of *AMPK-α*, a key regulator of the energy balance in cancer growth, as well as apoptotic markers such as *p53* and *caspase-3*, along with autophagic markers including *beclin1* and *ATG4A*. This combination therapy of metformin and paclitaxel exhibited significant potential as a treatment option for HepG2 liver cancer. In summary, the combination of metformin and paclitaxel not only enhances treatment efficacy but also reduces side effects. It induces cell cycle alterations and apoptosis and modulates key cellular proteins involved in cancer growth, making it a promising therapy for HepG2 liver cancer.

## 1. Introduction

Chemotherapy can, among other serious side effects, impair bone marrow function and cause gastrointestinal problems. If the patient develops resistance to this medication, their condition may deteriorate [1]. Certain drugs like as metformin (MET) can be used regularly or in conjunction with standard chemotherapy treatments to reduce side effects and increase the efficacy of the therapies [2].

Metformin is a popular antidiabetic drug for diabetics and obese people. Metformin has previously been shown in studies to have anticancer effects. Individuals with type 2 diabetes who took metformin had a lower risk of developing cancer [3]. Paclitaxel (PXT), a taxine model, was isolated from the bark of a coastal plant, the Pacific yew tree. Sarcomas, epithelial malignancies, and melanomas are all treated with it. Paclitaxel can also be used to treat malignancies of the breast, bladder, endometrium, non-small cell lung, and cervix [4]. Further study has revealed that MET inhibits cancer cell proliferation while also inducing apoptosis, autophagy, and cell cycle arrest in vivo and in vitro. The findings show that MET could be utilized as an adjuvant therapy for cancer as a repurposed drug [5,6].

Several biological substances have been shown to activate or boost the activity of the p53 tumor suppressor, resulting in cell cycle arrest, apoptosis cascades, and apoptosis induction. There is evidence linking p53 haploinsufficiency, increased mTOR signaling, and aggressive HCC. As a result, blocking mTOR could be a viable therapy strategy for aggressive HCC by lowering tumor-promoting activity caused by p53 haploinsufficiency [7].

Autophagy has a disputed role in cancer genesis, depending on tumor stage and genetics. Autophagy was discovered in normal cells as an antitumorigenic function by isolating damaged organelles. To proliferate swiftly in the early stages, cancer cells require high basal levels of autophagy. However, autophagy is linked to hypoxic regions in advanced tumors with higher metabolic needs. Autophagy effectively regulates energy levels in advanced cancers, recycles intracellular components, and assembles tumor-promoting factors [8]. Targeting the mTORC1 pathway could be an important breakthrough in the development of medicines for chronic diseases and cancer. While the most effective mTORC1 inhibitors showed promising results in preclinical trials, they also have substantial limitations, such as nonspecific inhibition of mTORC1 and mTORC2, which may limit their utility [9].

The fundamental goal of anticancer drugs is to inhibit cell division while increasing apoptosis. As a result, p53 plays an important role as one of their main targets. Furthermore, it has been proposed that AMPK activation aids in the activation of p53. Furthermore, p21, a downstream element of the p53 cascade, is important in boosting p53-mediated tumor suppression in response to DNA damage by triggering G1/S arrest [7].

Raising the dose of chemotherapy drugs to improve treatment for cancer may increase overall toxicity and damage. Alternatively, by combining them with less toxic medicines, this problem can be eased. Combining the chemotherapy drugs PXT and MET may improve therapeutic efficacy while decreasing negative effects. The use of MET as an adjuvant medicine with PXT in human cell lines for hepatocellular carcinoma, colon cancer, and breast cancer was explored. This study investigated the cell cycle, antiproliferation biomarkers, apoptosis, and autophagy.

## 2. Materials and Methods

### 2.1. Evaluation of Antitumor Effects

HepG2, HCT116, and MCF-7 cells were obtained from the Tissue Culture Unit, Department of Biochemistry, Faculty of Science, King Abdulaziz University. The antitumor efficacy of PXT and MET and their combination were investigated on HepG2, HCT116, and MCF-7 cells. MCF-7 cells were cultured in RPMI-1640 medium supplemented with 10% inactivated serum and 100 units/mL penicillin/streptomycin. HepG2 and HCT116 cells were grown in DMEM medium. The cells were incubated in 5% CO_2_ at 37 °C until they were 80–90% confluent. The cells were harvested and collected using 0.25% trypsin-EDTA before being stained and counted with 0.4% trypan blue. Following that, 10,000 HepG-2, MCF-7, and HCT116 cells were grown in 96-well microplates and incubated at 37 °C in a CO_2_ incubator for adhesion overnight. The cells were subjected to three distinct treatments, each with a different dose of metformin (0–1000 μg/mL), paclitaxel (0–100 μg/mL), or a combination of the two. The microplate was incubated for 24–48 h at 37 °C. An amount of 100 μL of medium containing 0.5 mg/mL MTT (provided by Gold Bio, St. Louis, MO, USA) was replaced in each well, and samples were then incubated at 37 °C for 4 h before the medium was changed out for 100 μL of dimethyl sulfoxide (DMSO) (supplied by Gold Bio). Absorbance was measured at 540 nm (Bio-RAD microplate reader, Hercules, CA, USA). Percentage of viability was then computed [10,11].

### 2.2. Assessing the Apoptosis in Treated HepG2 Cells Using Annexin V-PE/7-AAD

HepG2 cells were kept alive in a CO_2_ incubator for 24 h after transplanting before being scraped and counted. After 24 h, 2 × 10^5^ cells were cultivated in each well of a 6-well plate, and the medium was changed with one containing the IC_50_ concentrations of PXT, MET, and a combination of both at 500 μg/mL MET and the IC_50_ concentration of PXT. The cells were scraped after 24 h, the medium was removed, and the pellets were centrifuged. The cells were then washed with phosphate buffer solution (PBS) and suspended in it. PBS was used to suspend the cells. A microplate (96 wells) was filled with 200 μL of suspended HepG-2 cells that had been treated with a drug, as well as 30 μL of the Gava Nexin reagent. The plate was incubated in the dark for 20 min. The fluorescence of Annexin V-PE and 7-AAD was identified using a flow cytometry technique. The annexin V-PE/7-AAD dye was purchased from Guava Technologies, Inc. The software module performs computations automatically [12].

### 2.3. Flow Cytometric Assessment of Cell Cycle Progression in Treated HepG2 Cells Using the Propidium Iodide

Propidium iodide (PI) (supplied by Thermo-Fisher Scientific, Waltham, MA, USA) can be used to stain and bind DNA as well as evaluate cellular aggregation across the cell cycle using flow cytometry [13]. In each well of a 6-well plate, 1 × 10^6^ HepG2 cells were grown for 24 h. The medium was replaced with a medium containing an IC_50_ dose of PXT, IC_50_ of MET, or both IC_50_ of PXT and 500 μg/mL MET. Cells were scraped and washed twice with PBS after a 24 h incubation period. After the suspend cells were centrifuged at 1500 rpm for 5 min, they were fixed with 1 mL of 70% ethanol to fix the cells. The 100 μL sample of suspended cells was stained for an hour in the dark with 250 μL of PI solution (50 μg/mL PI). A flow cytometer (Applied Biosystems, Waltham, MA, USA) was used to examine all labelled cells [13].

### 2.4. Molecular Docking Study: Ligand and Protein Determination and Preparation

The computational experiments were conducted using a molecular docking server (https://www.dockingserver.com accessed on 13 November 2021 (r306)) that is based on AutoDock 4 for docking calculation [14]. The partial charges of the protein and ligand were calculated using the PM6 technique using the MOPAC2009 program [15]. The necessary hydrogen atoms and solvation parameters were added using AutoDock tools. The van der Waals and electrostatic energies were calculated using distance-dependent dielectric functions. Docking simulations were performed using the Solis and Wets methods, the Lamarckian genetic algorithm, and other methods. Each docking test was explained using data from ten independent runs, each of which was designed to end after receiving 250,000 maximum energy evaluations.

Using the PubChem database, the three-dimensional (3D) chemical structures of the selected molecules of proteins interacting with MET and PXT were retrieved. The 3D and geometry improvements with ligand energy reduction were performed using algorithms controlled by the docking server. The built-in Merck Molecular Force Field94 (MMFF94) was utilized for the geometry optimization strategy for the ligand synthesis module, and the charge calculation method Gasteiger was applied at pH 7.

AMPK, autophagy, and apoptotic proteins were picked from the RCSB Protein Data Bank (http://www.rscb.org accessed on 5 July 2023). The molecules that medications were made to block were protein kinases, specifically AMPK core with ATP (PDB: 4EAG), EGFRK (PDB: 1M17), and FKBP12-rapamycin-mTOR-FRB (PDB: 1FAB), the PPIase domain of FKBP51-mTOR-FRB (PDB: 4DRH). Other important autophagy proteins included Beclin 1 (PDB: 3Q8T), LC3 (structure of the WD40 domain of human ATG16L1), (PDB: 5NUV), and ATG4A (PBB 2P82) as autophagic proteins. In addition to apoptotic genes, TP53 (PDB: 3DCY) and caspase-3 (PDB: 3GJQ) were included.

The protein structures were generated using the protein preparation wizard in the molecular docking server panel. Bond ordering was formed, and hydrogen atoms were also introduced. Water molecules and other unimportant substances were eliminated. Affinity (grid) maps with a 202,020 grid and 0.375 spacing were made using the Auto grid application [16]. The AutoDock parameter set- and distance-dependent dielectric functions were used to determine the van der Waals and electrostatic components, respectively.

To simulate docking, the Solis and Wets local search method and the Lamarckian genetic algorithm (LGA) were also employed [17]. The initial locations, orientations, and torsion angles of the ligand molecules were determined at random. Each docking experiment consisted of 100 separate runs, each of which was designed to terminate after a maximum of 2,500,000 energy evaluations. The search was performed with a translational step of 0.2, a quaternion step of 5, and a torsion step of 5. After each docking computation, the root means square deviation (RMSD) between the docked ligand pose with the lowest energy and the ligand pose with the most complex crystal structure was computed. The stance with the smallest RMSD was picked as the pose out of all the stances the docking program conducted.

The molecular docking server generated the estimated free energy of binding (kcal/mol) as G values. They were then converted into the predicted inhibition constants (Ki). For the analyzed docking poses, the Ki values were derived using the G parameters [18]:ΔG = RT (lnKi)
$$\mathrm{Ki}\;=\;e^{\frac{\Delta G}{RT}}$$
where R (gas constant) is 1.98 cal (mol K)^−1^ and T (room temperature) is 298.15 K. The complexes were looked at using the web tool Protein–Ligand Interaction Profiler (PLIP) following docking (Technical University of Dresden) [19].

### 2.5. The Gene Expression of Adenosine Monophosphate Kinase (AMPK-α), Epidermal Growth Factor (EGFR), Total p53 (Tp53), Caspase-3, Beclin 1, and ATG4A

#### 2.5.1. Extraction of RNA

After being subjected to IC_50_ PXT, IC_50_ MET, or both IC_50_ of PXT and 500 μg/mL MET, HepG-2 cells were harvested as previously prepared for the cell cycle investigation. HepG2 cells were treated with the lysis buffer from an RNA Purification Kit to extract pure RNA (QUIAGEN Kit, Saxony, Germany). Guanidine-thiocyanate was present in every utilized solution, which was also RNAase free. Cells were homogenized, and the full homogenate was added to a micro spin column and thoroughly mixed with 600 μL of 70% ethanol. After that, RNA was extracted from the micro spin column by eluted solution. Eluted high-quality RNA was collected, and RNA was measured by nanodrop microvolumes (30–100 μg/mL).

#### 2.5.2. cDNA Synthesis

Revert Aid H minus reverse transcription kits (Thermo Scientific, Fermentas, Waltham, MA, USA, # EP0451) were used to convert RNA into complementary DNA for cDNA synthesis (cDNA). On the cool side of a sterile, nuclease-free tube, 5 μg of RNA template was added, then 0.5 μg of Oligo dT in 12.5 μL of DEPC-treated water. An amount of 6.6 μL of downstream and upstream primers, 2 μL of dNTPs, buffer, 1 μL of reverse transcription, 8 μL of DNA Taq polymerase (5 units), 3.2 μL of 2 mM magnesium chloride, and 100 μL of nuclease-free water were added to the mixture. At 42 °C, it was incubated for 60 min. After heating the mixture for 10 min at 70 °C, it was finished.

#### 2.5.3. Real-Time qPCR with SYBR Green

Real-time quantitative polymerase chain reaction with SYBR Green was used to measure gene mRNA expression in treated HepG-2 cells, with GAPDH serving as an internal reference (PCR). A SYBR Green/ROX qPCR Master (Thermo Scientific, USA, # K0221) and gene primers were used. Adenosine Monophosphate Kinase (AMPK-α), Epidermal Growth Factor (EGFR), Total P53 (Tp53), Caspase3, Beclin1, and ATG4A primers are shown in Table 1. The primers were chosen using the website (http://www-genome.wi.mit.edu/cgi-bin/primer/primer3www.cgi website of the Whitehead Institute for Biomedical research that accessed on 13 October 2022) and double-checked and validated as unique via similarity checking using the Basic Local Alignment Search Tool (BLAST) (www.ncbi.nlm.nih.gov/blast/Blast.cgi, accessed on 13 October 2022).

A 3 μL (10–20 ng/L) cDNA template, 2 μL (forward and reverse) of the target gene, up to 12.5 μL of SYBR Green, and 25 μL of water are required for the polymerase chain reaction. StepOnePlus (Applied Biosystems, Life Technology, Carlsbad, CA, USA) placed the finished reaction mixture into a real-time heating cycle. Each PCR cycle consisted of initial denaturation (95 °C/10 min), annealing, and extension of the specific gene expressed (95 °C/15 s, 60 °C/30 s, and 72 °C/30 s, respectively). The number of the cycle was 40.

At the end of the preceding cycle, the temperature was increased by between 60 and 95 °C to reach the melting curve. An example of the so-called housekeeping gene (GAPDH) was used to calculate the relative gene expression or fold change in the target gene. The critical quantities (Ct) of the target genes were normalized using the quantities (Ct) of the housekeeping gene by 2^−∆∆Ct^ [20].

### 2.6. Statistical Analysis

The viability of treated cells is given as a mean ± standard deviation (SD). By multiplying the absorbance of treated cells by 100 and dividing it by the absorbance of untreated cells, the percentage of viability was estimated. GraphPad Prism Software (version 9.0, San Diego, CA, USA) was used to calculate mean and SD and compute the drug IC_50_. The flow cytometry software from Applied Biosystems automatically calculated the proportion of cells in each phase of the cell cycle as well as the percentage of cells of necrotic and apoptotic cells.

## 3. Results

### 3.1. Antitumor Effects of Paclitaxel and Metformin and Their Association on HepG2, HCT116, and MCF-7 Cells

In HepG2, HCT116, and MCF-7 cell lines, the IC_50_ values for MET were 385.3, 296, and 279 μg/mL, respectively. When compared to PXT alone, the combined IC_50_ of PXT and MET decreased HepG2, HCT116, and MCF-7 cell viability by 1.8, 1.28, and 1.44 times, respectively (Figure 1 and Table 2).

The IC_50_ values of PXT-treated HepG2 cells were 1.5 times lower than the IC_50_ values of PXT-treated HCT116 cells and 3.7 times lower than the IC_50_ values of PXT-treated MCF-7 cells. Additionally, the IC_50_ values of the combined PXT and MET treatment were 2.1 times lower than the IC_50_ values of the combined PXT and MET treatment on HCT116 cells and 4.8 times lower than the IC_50_ values of the combined PXT and MET treatment on MCF-7 cells. The most significant effects of PXT and the combined PXT and MET treatments were observed in HepG2 cells compared to HCT116 and MCF-7 cells. As a result, experiments that assessed the apoptosis, cell cycle, and gene expression of apoptotic and autophagic proteins were performed on HepG2 cells exclusively.

### 3.2. Percentage of Apoptosis in HepG2 Treated Cells

The combination of PXT and MET exhibited the highest level of early apoptotic activity (21.2%) in HepG2 cell lines. Initial apoptosis detection using annexin V-PE/7-AAD dye revealed that the early apoptotic activity in HepG2 cell lines treated with the combination of PXT and MET was 2.2 to 4 times greater than that observed in cells treated with either paclitaxel alone or metformin alone (Figure 2 and Figure 3).

### 3.3. Flow Cytometry Analysis of the Cell Cycle

There were no significant changes between untreated HepG2 cells that stopped the cell cycle in G0/G1 (61%), S phase (12%), and G2/M (16%) and treated HepG2 cells that received 500 μg/mL of IC_50_ MET alone in the G0/G1 phase (55%), S phase (13%), and G2/M phase (13%). In the subG0 phase of the cell cycle, MET slightly increased the accumulation of apoptotic cells (Figure 4 and Figure 5).

HepG2 cells treated with IC_50_ PXT alone at a dosage of 10 μg/mL showed a significant increase in G2/M phases (26%) in contrast to untreated HepG2 cells, which demonstrated cell cycle arrest in S phase (12%) and G2/M (16%). HepG2 cells arrested in G0/G1 phase treated with PXT (64.4%) did not differ significantly from untreated HepG2 cells arrested in G0/G1 phase (61%) (Figure 4 and Figure 5).

The percentage of cells in phase G1/G0 of HepG2 treated with combination PXT and MET decreased significantly when compared to HepG2 treated with PXT or MET alone; however, the percentage of cells in S phase and G2/M phases increased in HepG2 treated with combined PXT and MET (Figure 4 and Figure 5).

### 3.4. Metformin and Paclitaxel Bind to Related Protein Signaling Pathway

AMPK modulation (downstream and/or upstream regulatory proteins of cell signaling pathways involved in cell survival and proliferation) is the biological mechanism by which PXT and MET modulate the activity of apoptosis and autophagy. The complexes were docked, and the PLIP web server was used to evaluate them.

The possible docking modes of protein–ligand interaction were calculated and displayed by the molecular docking server based on AutoDock 4. This enables a molecular understanding of the interaction between PXT and MET 3D structures imported from the PubChem database and the crystal structures of protein kinases (AMPK, EGFRK (PDB ID: 4EAG), 1M17); autophagy-related proteins such as FKBP12-rapamycin-mTOR-FRB, the PPIase domain of FKBP51-mTOR-FRB, Beclin 1, and ATG16L1 ATG4A (PDB ID: 1FAP, 4DRH, 3Q8T, 5NUV, 2P82); and apoptosis-related proteins (TP53, Caspase3, (PDB ID: 3DCY, 3GJQ)), which were imported from the RCSB protein data bank.

We were able to pinpoint the precise amino acid residues directly engaged in the MET or PXT–AMPK interaction. Figure 6A shows MET binding to the active residues in AMPK’s active site, LYS–126, MET–84, ARG–117, and GLN–122, through hydrogen bonding and THR–86, ARG–117, GLN–122, and ASP–89 through hydrophobic bonding, with a total binding energy of −2.76 kcal/mol. PXT binds to ASP-89(2), GLN-122, and ARG-223 by hydrogen bonds, as well as to THR-88, TYR-120, GLN-122, LYS-126, VAL-129(2), ILE-149(2), HIS-150, and ARG-223 through hydrophobic bonds, as shown in Figure 1A, with a total binding energy of −11.01 kcal/mol (Table 3 and Table 4). These binding energy estimates attest to the strong binding of PXT and weak binding of MET to the active residues of AMPK. The screening study is anticipated to reveal the mechanism by which PXT suppresses AMPK rather than MET.

Figure 6B displays MET binding to the active residues at the EGFRK active site: CYS-773(2), PHE-771, and TYR-777 through hydrogen bonds and LEU-694, PHE-699(2), VAL-702(3), ALA-719, ARG-817, and LEU820 through hydrophobic bonds, with a total binding energy of about −4.92 kcal/mol. Figure 6B demonstrates that PXT attaches to ASP-776 through hydrogen bonding and to THR-88, TYR-120, GLN-122, LYS-126, VAL-129(2), ILE-149(2), HIS-150, and ARG-223 through hydrophobic bonding, with a total binding energy of approximately −10.59 kcal/mol (Table 3 and Table 4).

In order to find a compound that will allosterically block mTORC1 in this study, an in silico technique was used in light of the structural accessibility of FKBP12 RAPFRB. Small molecules that might take the place of rapamycin in the binding pocket at the FKBP12/FRB interface were the focus of our search. Smaller virtual ligands might fit in the rather large binding pocket for rapamycin.

MET binds to the active residues at FKBP12/LYS-47(2), FRB’s GLU-54, TYR-210(2), and GLU-54(2) by hydrophobic bonding and hydrogen bonding, respectively, with a total binding energy of −4.76 kcal/mol. PXT binds to TYR-82 and GLU-54 through hydrogen bonds, and to TYR-26, PHE-46, VAL-55, ILE-56, TRP-59, TYR-82, ILE-90, PHE-99, PHE-2039(2), THR-2098, TYR2-105, HIS-8, SER-203, and ASP-37 through hydrophobic bonds, with a total binding energy of−15.63 kcal/mol (Table 3 and Table 4).

The active residues at FKBP55- mTOR GLY-117, LEU-119, PHE-203, SER-118, and TYR-203 are shown to bind to MET by hydrogen bonds, whereas ARG-204, ALA-116, and TYR-2038 are shown to bind via hydrophobic bonds, with a total binding energy of around −4.31 kcal/mol. PXT binds to bind to TYR-57, GLN-85, and SER-2035 through hydrogen bonding and PHE-67, PHE-77, GLN-85, VAL-86, LYS-121, ILE-122, GLU-2032, ARG-2036, PHE-2039(2), TRP-2101, TYR-2105, TRP-90(2), and ARG-2036 through hydrophobic bonding, with a total binding energy of −7.33 kcal/mol (Table 3 and Table 4).

The active residues of Beclin 1 GLU-224, ASPB-221, and ALA-217 are bound by MET by hydrogen bonds, whereas LEU220 and ASPA221(2) are bound by hydrophobic bonds, with a total binding energy of about -7.41 kcal/mol Figure 6C. PXT binds to GLU-216A, GLU-216B, ARGA-219(2), LEU-220, ALA-215, LYS-212, GLU-216A (2), GLU-216(B), GLU-223, ARG-219A, and ARG-219B through hydrophobic bonding, as shown in Figure 6C, with a total binding energy of −6.92 kcal/mol (Table 2 and Table 3).

MET binds to the active residues at ATG16L1d’s CYS-356 and GLU-357 by hydrogen bonding and PHE-318, PHE-358, TRP-349, and CYS-316 by hydrophobic bonding, with a total binding energy of about −4.31 kcal/mol. PXT binds to ARG-345 and LYS-347 via hydrogen bonds, as well as to PHE-318(3), ASP-319, TRP-349(2), PHE-358 (2), and GLU-357 via hydrophobic bonds, with a total binding energy of approximately −7.33 kcal/mol. (Table 2 and Table 3).

The active ATG4A residues at ASP-174(2), SER-172, ASN-175(2), and PRO-228 are shown to bind to MET via hydrogen bonds and hydrophobic bonds, respectively, with a total binding energy of approximately −5.27 kcal/mol. PXT binds to ASN-175 by hydrogen bonds and PRO-228, ARG-230, and ILE-233(2) through hydrophobic bonds, with a total binding energy of about −8.85 kcal/mol (Table 3 and Table 4).

Figure 6D demonstrates that MET binds to the active residues at TP53′s PRO-30, LEU-31(2), and GLU-29 via hydrogen bonds with a total binding energy of −3.25 kcal/mol. According to Figure 6D, PXT forms hydrogen bonds with ARG-15, PHE-36, SER-32, and GLU-33 and hydrophobic bonds with ASP-28,85, LEU-59, LYS-63, GLN-64, HIS-67, TYR-83, TYR-83, LYS-63, and LYS-63. The overall binding energy is −6.04. kcal/mol (Table 3 and Table 4).

The active residues at caspase3′s SER-63, SER-65(3), THR-62, and HIS-121 are shown to bind to MET by hydrogen bonds and ARG-64 via hydrophobic bonds, respectively, with a total binding energy of approximately −3.78 kcal/mol (Figure 6E). PXT binds to THR-62(2) and SER-65 through hydrogen bonds, and to ARG-64, CYS-163, HIS-121, SER-65, ASP-70, and GLN-161 through hydrophobic bonds, as shown in Figure 6E, with a total binding energy of around −8.42 kcal/mol (Table 3 and Table 4).

### 3.5. Genes Expression of AMPK-α, EGFR, TP53, Caspase-3, Beclin 1, and ATG4A in Treated HepG2

The relative mRNA expressions of *AMPK-α, EGFR, TP53, Caspase-3, Beclin 1,* and *ATG4A* were evaluated inHepG2 cells treated with MET, PXT, or PXT combined with MET.

The expression of AMPKa was upregulated in HepG2 cells treated with MET and combined PXT and MET 5.57- and 2.92-folds, respectively; it was downregulated 0.6-fold in HepG2 cells treated with PXT. AMPK-a gene expression was upregulated in HepG2 cells treated with combined PXT and MET compared to HepG2 cells treated with PXT but downregulated by 1.9-fold compared to HepG2 cells treated with MET. EGFR expression increased 0.936-fold, 3.82-fold, and 8.44-fold in HepG2 cells treated with MET, PXT, or combined PXT and MET, respectively. HepG2 cells treated with combined PXT and MET increased the expression of the EGFR gene compared to HepG2 cells treated with PXT alone and compared to HepG2 cells treated with MET alone. Tp53 expression increased 5.41-fold, 4.568-fold, or 7.99-fold in HepG2 cells exposed to MET, PXT, or combined PXT and MET, respectively. In HepG2 cells treated with combined PXT and MET, Tp53 gene expression was upregulated as compared to HepG2 cells treated with PXT and HepG2 cells treated with MET. In HepG2 cells treated with MET, PXT, or PXT plus MET, the expression of caspase3 increased 3.6-fold, 66.5-fold, and 33.0-fold, respectively. In HepG2 cells treated with paclitaxel and metformin, caspase-3 gene expression was upregulated more than in cells treated with MET and downregulated relative to cells treated with PXT alone. Beclin1 expression increased 10.7, 4.26, or 12.1 times, respectively, in HepG2 cells treated with MET, PXT, or in combined PXT and MET. Beclin1 gene expression increased in HepG2 cells treated with PXT plus MET compared to HepG2 cells treated with PXT and compared to HepG2 cells treated with MET. ATG4A expression increased 11.06-fold, 1.77-fold, or 11.98-fold in HepG2 cells treated with MET, PXT, or PXT plus MET, respectively. ATG4A gene expression was elevated in HepG2 cells treated with combined PXT and MET compared to those treated with PXT alone and compared to those treated with MET alone (Table 5 and Figure 7 and Figure 8).

## 4. Discussion

Metformin inhibits inflammation, development of new blood vessels, and invasion of malignant cells. Metformin has been proven to cause apoptosis in a variety of cancers, including breast, prostate, and liver cancers [21,22]. The effectiveness of PXT therapy alone was compared to its effectiveness when paired with MET in the current study. When PXT was combined with MET, the IC_50_ was reduced by 1.8-fold, 1.27-fold, and 1.44-fold in the viability of HepG2, HCT116, and MCF-7 cells, respectively, compared to PXT therapy alone.

Recent research reveals that combining PXT, cisplatin, or doxorubicin with MET improves chemotherapeutic efficacy in reducing breast tumor growth. Both MET and PXT could suppress MCF-7 and A549 cells, but the combination of both therapies was found to be more effective than each therapy alone [23]. PXT, a drug that inhibits the formation of endometrial cancer cells, was coupled with MET to improve endometrial cell growth inhibition [24].

According to the findings, utilizing annexin V in combination with IC_50_ PXT and MET boosted the early stage of HepG2 cell line apoptosis to 21%, which was higher than cells treated with IC_50_ PXT (9.2%) or MET (5.3%) alone. Early apoptosis was also shown to be 2.3 or 4 times more common in HepG2 cell lines treated with IC50 PXT plus MET than in those treated with IC50 PXT or MET alone. Furthermore, MET has been shown to trigger apoptosis in colon, prostate, and breast cancer cells [25].

In the current study, it was found that treatment with PXT alone or in combination with MET increased the production of cleaved caspase 3, a key protein involved in the execution of apoptosis. Paclitaxel has previously been reported to induce apoptosis in HepG2 cells by activating caspase 3. Caspase-3 activation in HepG2 cells is thus thought to play a role in the PXT-induced apoptotic pathway. These findings are consistent with previous research [26].

The current study found that administering PXT alone or in conjunction with MET arrested HepG2 cells in the G1 phase (64% and 32%, respectively), the S phase (12% and 26%), and the G2/M phase (16% and 34%). The higher percentage of cells halted in the G2/M phase shows that this medicine can decrease neoplastic progression and boost early apoptosis in tumor-bearing mice [23]. Previous studies showed that PXT-treated epithelial osteosarcoma (U-2 OS) and non-small cell lung cancer (NSCLC) cells accumulate in the G2/M phase and thereafter undergo apoptosis via caspase-3 activation [27,28,29].

According to research, PXT can bind to a subunit of tubulin, preventing microtubule depolymerization. Microtubule stabilization is critical in regulating cell division throughout the M phase of the cell cycle, inhibiting malignant cell development during the G2/M phase, and ultimately leading to cell death [27]. Cells halted in the S phase of the cell cycle were shown to increase apoptosis in triple-negative breast and pancreatic cancer cell lines treated with MET. Metformin inhibited DNA synthesis and halted the cell cycle in the S phase in hepatocellular carcinoma cells, resulting in cell death via the AMPK-α dependent pathway [30].

AMPK-α expression increased 5.57-fold and 2.92-fold in HepG2 cells treated with MET or MET plus PXT, respectively. In HepG2 cells treated with paclitaxel plus metformin, AMPK-α gene expression was elevated 4.86 times more than in HepG2 cells treated with paclitaxel alone.

MET was found in several studies to decrease ATP and increase AMP in the inner mitochondrial membrane by inhibiting NADH (Complex I) dehydrogenase. AMPK, which enhances fatty acid oxidation and inhibits hepatic gluconeogenesis, is connected to AMP. AMPK-α inhibits the mammalian target of rapamycin (mTOR). Tyrosine and serine/threonine are phosphorylated by the protein kinase mTOR. When MET suppresses the mTOR signal, a cascade of proteins that control growth, reproduction, motility, cell survival, and protein synthesis is formed. Increased mTOR signaling activity has been associated with prostate, brain, lung, bladder, and kidney cancers. The epidermal growth factor (EGF) is thought to be downstream of the mTOR pathway [31]. Another research study indicated that the treatment of SK-MEL-28 melanoma cells with PXT and MET for 24 h resulted in the activation of AMPK and the mTOR pathway downstream [32].

According to this study, the impact of PXT, MET, and the combination of PXT and MET on EGFR expression in HepG2 cells were 3.8, 3.6, and 7.07 times greater, respectively. Previously, it has been shown that PXT prompts the death of squamous cells in human oral cancer cells by obstructing the EGFR signaling pathway [33].

The EGFR signaling system comprises multiple pathways, with the critical ones being p38, ERK1/2, JNK, and MAPK. The ERK1/2 molecule plays a pivotal role in EGFR downstream signal transmission [34]. Once activated, ERK1 and ERK2 translocate to the nucleus, thereby increasing the expression of certain oncogenes [35]. In contrast, active JNK and p38 were found to decrease apoptotic activity and suppress cell apoptosis by obstructing caspase-3 [36].

The p53 protein protects cell division and triggers programmed cell death [37]. When HepG2 cells were treated with MET, PXT, or a combination of the two, p53 expression increased 5, 4.5, and 8 times, respectively. Paclitaxel induces apoptosis in nasopharyngeal cancer cells, which is heavily dependent on the p53 pathway [37]. Paclitaxel destroys the mitotic spindle in noncancerous fibroblasts, preventing malignant cells from dividing. Paclitaxel disrupts microtubules, potentially via the p53 pathway, resulting in cell cycle arrest and death [38]. Metformin regulates the p53 gene, which inhibits cancer cell development and produces programmed cell death. Metformin activates AMPK-α, which reduces mTOR and increases p53 in hepatocellular carcinoma through enhancing p53 phosphorylation on serine 15. This causes the cell cycle to halt and the proliferation of HCC cells to decrease [39].

In HepG2 cells treated with MET, PXT, or PXT plus MET as compared to untreated HepG2 cells, caspase-3 expression was increased. Paclitaxel blocks the G2/M cycle and triggers cell death in human osteosarcoma U-2 Os cells by activating the caspase-3 pathway. Caspase-3 is responsible for chromosomal DNA fragmentation, extrinsic apoptosis, and PARP-induced cleavage. It binds to Apaf-1 and activates procaspase-9, which subsequently causes caspase-3 to be activated through the intrinsic pathway [40]. In breast cancer cells, metformin stimulates caspase3, which boosts caspase-8 and results in apoptosis [41].

Caspase-3 expression increased in HepG2 cells treated with MET, PXT, or PXT combined with MET, when compared to untreated HepG2 cells. Paclitaxel suppresses the G2/M cycle and induces cell death in human osteosarcoma U-2 Os cells by activating the caspase-3 pathway. Caspase-3 is the enzyme responsible for chromosomal DNA fragmentation, PARP-induced cleavage, and extrinsic apoptosis. It binds to Apaf-1 and promotes procaspase-9, which activates caspase-3 via the intrinsic pathway [40]. Metformin activates caspase-3, which then activates caspase-8 and causes apoptosis in breast cancer cells [41].

Autophagy can either promote or inhibit cell death depending on the specific conditions [42]. ATG4A, a cysteine protease, can interact with GABARAP, the γ aminobutyric acid receptor-associated protein. GABARAP possesses a positively charged N-terminus and can bind to light chain-3 (LC3). ATG4A cleaves proLC3 to create LC3, exposing a glycine residue that permits it to bind to phosphatidyl-ethanolamine (PE) and become a component of autophagosome membranes. ATG4A is involved in autophagy-related processes including autophagosome formation and autophagic flux control [43].

In the present investigation, MET, PXT, and PXT combined with MET treated HepG2 cells, resulting in significant increases in beclin1 expression. ATG4A expression was increased in HepG2 cells treated with MET, PXT, or PXT coupled with MET, with fold increments.

This study indicated that beclin1, a protein involved in regulating apoptosis and autophagy, played a role in these processes. In the presence of paclitaxel treatment, beclin1 interacted with an antiapoptotic Bcl-2, resulting in the formation of fragments that migrated to mitochondria and made the cells more susceptible to apoptotic signals [42,44]. The cleavage of ATG4D by caspase-3 led to the stimulation of autophagy. When fragmented ATG4D was overexpressed in cells, it induced a change in the membrane potential of the mitochondria, leading to the release of cytochrome c and the activation of intrinsic apoptotic pathways [45]. More research is needed to determine the precise molecular pathways by which caspase-3 fragments ATG4A, resulting in the activation of both apoptosis and autophagy.

Overexpression of beclin1 may affect the efficiency of PXT in preventing the growth of lung and cervical cancer cells [46,47]. By upregulating AMPK, which inhibits the mTOR pathway, MET has been shown to promote autophagy and cell death in a variety of cancer cells [48]. Stimulating autophagy could protect cells from cancer-causing proteins, which are thought to be the crucial source of tumor growth.

## 5. Conclusions

According to the current study, combining PXT with MET lowers the IC_50_ of PXT in HepG2, HCT116, and MCF-7, potentially lessening PXT’s deleterious effects. The arrest of the S and G2/M cell cycles resulted in early apoptosis in HepG2 cells treated with that mixture. The combination reduces mTOR signaling, which leads to cell death by boosting AMPK-α, which activates caspase-3, promotes EGFR, and activates p53. Inhibiting mTOR can result in beclin1 overexpression. Caspase-3 promotes ATG4A, and beclin1 fragments may trigger apoptosis. This study supports the efficacy of PXT and MET in the treatment of hepatocellular cancer. More research on the efficacy of anticancer drugs combined with MET in animals is urged as a novel strategy.

## Figures and Tables

**Figure 1 biomedicines-11-02688-f001:**
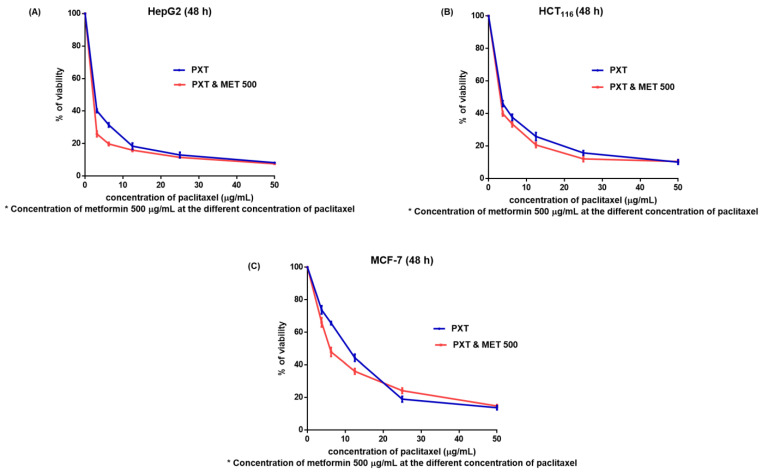
The viability of HepG2 (**A**), HCT116 (**B**), and MCF7 (**C**) cells (after 48 h) treated with various doses of paclitaxel alone or in combination with 500 μg/mL metformin (M).

**Figure 2 biomedicines-11-02688-f002:**
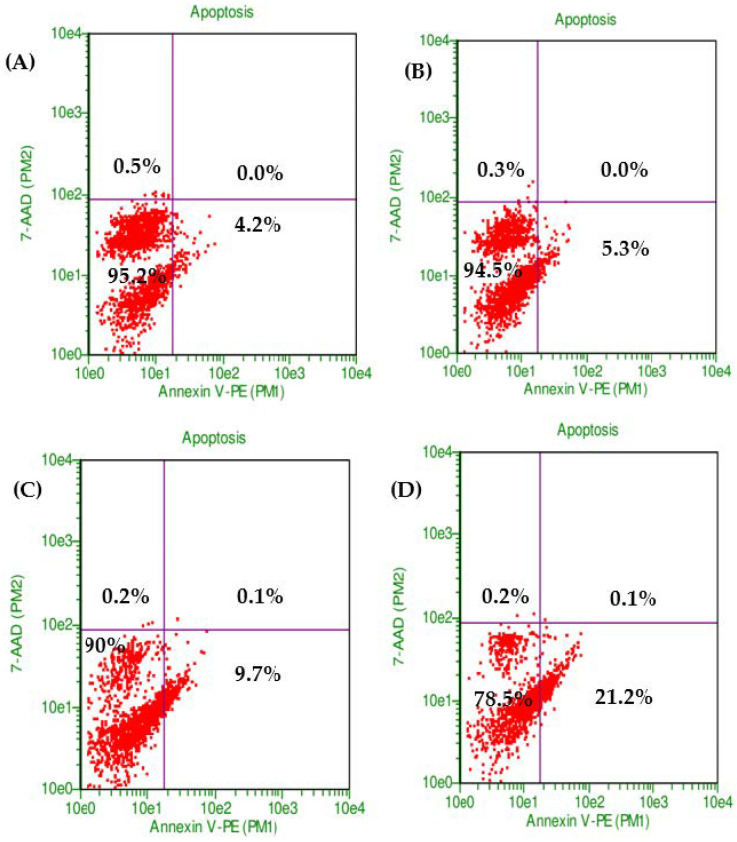
The proportion of viable cells and early stage of apoptosis in HepG2 without treatment (control) (**A**) or treated with MET (**B**), PXT (**C**), and PXT combined with MET (**D**), stained with annexin V and 7-ADD incubated 24 h.

**Figure 3 biomedicines-11-02688-f003:**
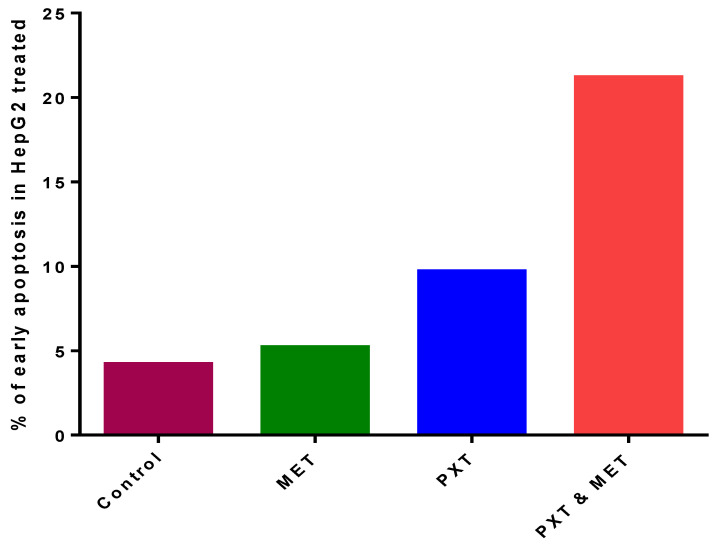
The percentage of early apoptosis in HepG2, treated with MET, PXT, and PXT combined with MET.

**Figure 4 biomedicines-11-02688-f004:**
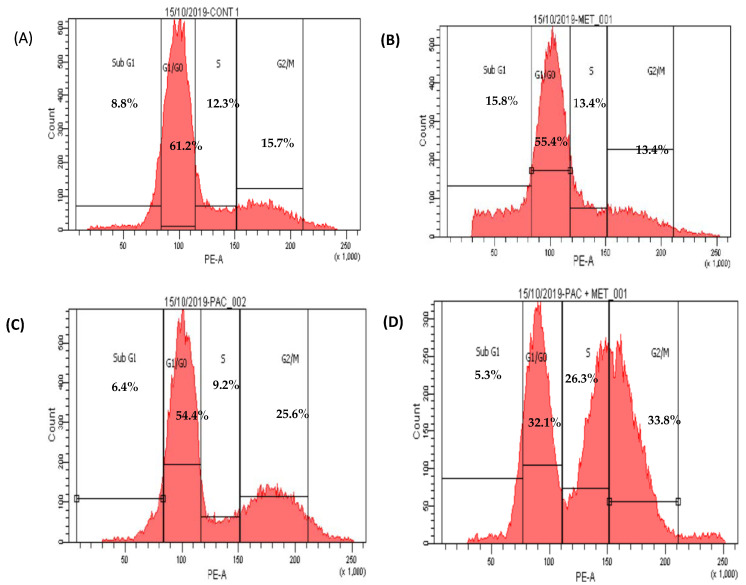
Scatter plots of cell cycle analysis stained by propidium iodide stain, cell cycle arrest of untreated HepG2 (**A**), MET (**B**), PXT (**C**), and PXT combined with MET (**D**). Chromatogram stages of cell cycle distribution were examined by flow cytometry.

**Figure 5 biomedicines-11-02688-f005:**
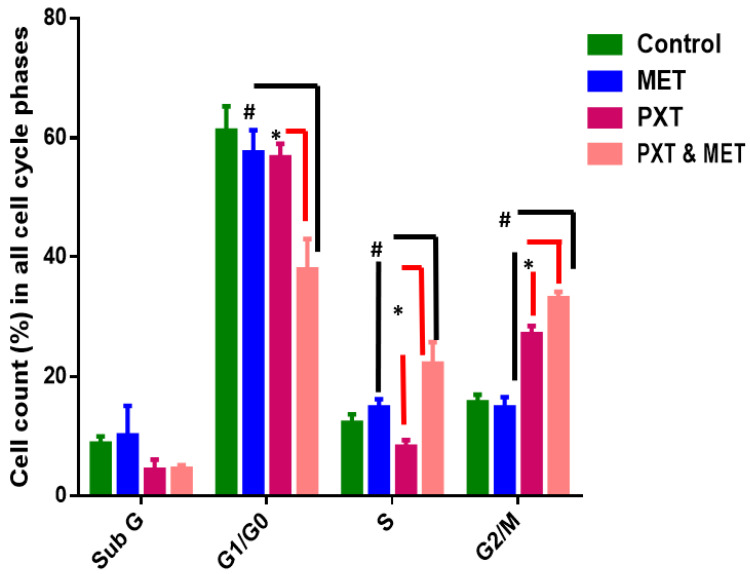
Percentage of cells in all cell cycle phases in HepG2 treated with MET, PXT, and PXT combined with MET. PXT induced significant G2/M phase arrest and decreased percentage of cells in the G0/G1 phase, and MET slightly induced the accumulation of apoptotic cells in the subG phase of the cell cycle. * significant differences between combined two drugs versus PXT, # significant differences between two combined drugs versus MET.

**Figure 6 biomedicines-11-02688-f006:**
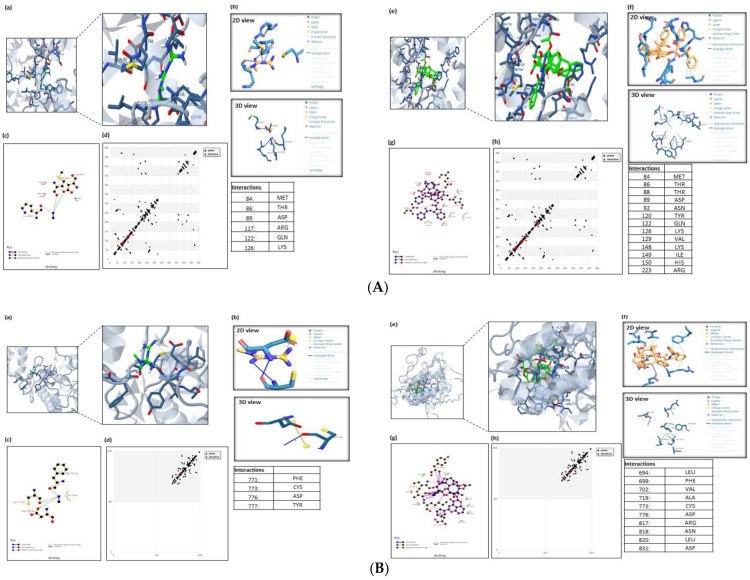
(**A**). Molecular docking analysis of AMPK with MET and PXT. The 3D modeling of the crystal structure of AMPK (PDB: 4EAG) with MET and PXT interacting with active residues of AMPK. Ligands docked with the target protein, generated using PLIP. The ligands used were b MET and f PXT. The 2D plot of the interaction of MET and PXT with active residues at AMPK’s active site (e.g., LYS-126, MET-84, ARG-117, GLN-122) and (e.g., ASP-89(2), GLN-122, ARG-223), respectively, through hydrogen bonding (HB) and with active residues at AMPK’s active site (e.g., THR-86, ARG-117, GLN-122, ASP-89) and (e.g., THR-88, TYR-120, GLN-122, LYS-126, VAL-129(2), ILE-149(2), HIS-150, ARG-223), respectively, through hydrophobic bonding. HB plot with active residues involved in AMPK–MET and PXT interactions. (**B**). Molecular docking analysis of EGFRK with MET and PXT. The 3D modeling of the crystal structure of EGFRK (PDB: 1M17) with MET and PXT interacting with active residues of EGFRK. Ligands docked with the target protein generated using PLIP. The ligands used were b MET and f PXT. The 2D plot of the interaction of MET and PXT with active residues at EGFRK’s active site (e.g., CYS-773(2), PHE-771, TYR-777) and (e.g., ASP-776), respectively, through hydrogen bonding (HB) and with active residues at EGFRK’s active site (e.g., LEU-694, PHE-699(2), VAL-702(3), ALA-719, ARG-817, LEU820) and (e.g., THR-88, TYR-120, GLN-122, LYS-126, VAL-129(2), ILE-149(2), HIS-150, ARG-223), respectively, through hydrophobic bonding. HB plot with active residues involved in EGFRK–MET and PXT interactions. (**C**). Molecular docking analysis of the TP53-induced glycolysis and apoptosis regulator protein with MET and PXT. The 3D modeling of the crystal structure of apoptosis regulator protein (PDB ID: 3DCY) with MET and PXT interacting with active residues of apoptosis regulator protein. Ligands docked with the target protein generated using PLIP. The ligands used were b MET and f PXT. The 2D plot of the interaction of MET with active residues at the apoptosis regulator protein’s active site (e.g., PRO-30, LEU-31(2), GLU-29) and (e.g., ARG-15, PHE-36, SER-32, GLU-33) through hydrogen bonding (HB) and hydrophobic bonding, respectively. The 2D plot of the interaction of PXT with active residues at the protein’s active site (e.g., ASP28,85, LEU59, LYS63, GLN64, HIS67, TYR83), respectively, through hydrophobic bonding. HB plot with active residues involved in TP53-induced apoptosis regulator protein–MET and PXT interactions. (**D**) Molecular docking analysis of caspase3 apoptosis protein with MET and PXT. The 3D modeling of the crystal structure of caspase3 protein (PDB ID: 3GJQ) with MET and PXT interacting with active residues of the caspase3 protein. Ligands docked with the target protein, generated using PLIP. The ligands used were MET and PXT. The 2D plot of the interaction of MET and PXT with active residues at the caspase3 apoptosis protein’s active site (e.g., SER-63, SER-65(3), THR-62, HIS-121) and (e.g., THR-62(2), SER-65), respectively, through hydrogen bonding (HB) and with active residues at caspase3 apoptosis protein’s active site (e.g., ARG-64) and (e.g., ARG-64, CYS-163, HIS-121, SER-65, ASP-70, GLN-161), respectively, through hydrophobic bonding. HB plot with active residues involved in caspase3 apoptosis protein–MET and PXT interactions. (**E**). Molecular docking analysis of Beclin 1, an essential autophagy protein, with MET and PXT. The 3D modeling of the crystal structure of Beclin 1 (PDB ID: 3Q8T) with MET and PXT interacting with active residues of Beclin 1 protein. Ligands docked with the target protein, generated using PLIP. The ligands used were b MET and f PXT. The 2D plot of the interaction of MET and PXT with active residues at the Beclin 1 protein’s active site (e.g., GLU-224, ASPB-221, ALA-217) and (e.g., GLU-216A, GLU-216B, ARGA-219(2)), respectively, through hydrogen bonding (HB) and with active residues at Beclin 1 protein’s active site (e.g., LEU220, ASPA221(2)) and (e.g., LEU-220, ALA-215, LYS-212, GLU-216A (2), GLU-216(B), GLU- 223, ARG-219A, ARG-219B), respectively, through hydrophobic bonding. HB plot with active residues involved in Beclin 1 protein–MET and PXT interactions.

**Figure 7 biomedicines-11-02688-f007:**
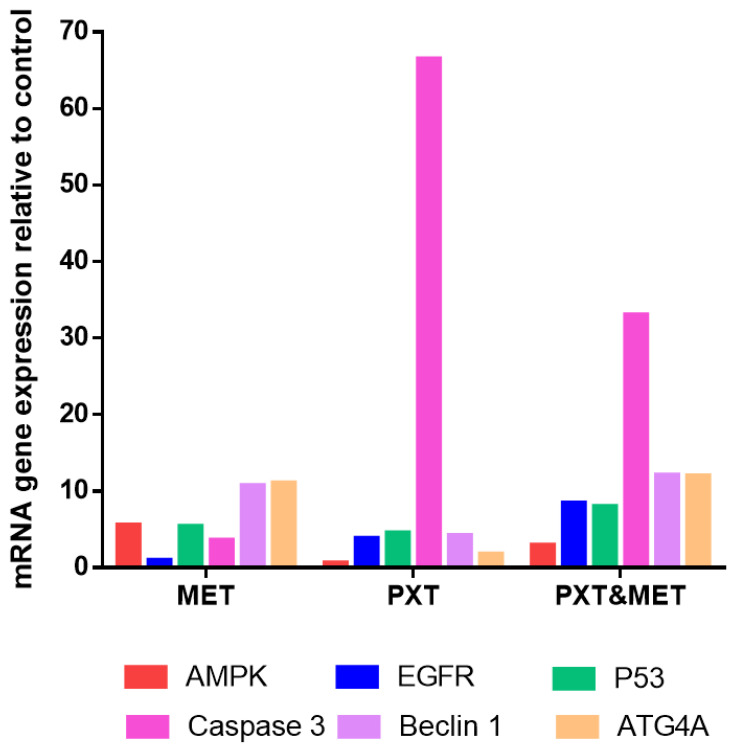
RT-qPCR mRNA expression relative to control of AMPK-α, EGFR, TP53, Caspase3, Beclin1, and ATG4A in HepG2 cells treated with MET, PXT, or PXT&MET.

**Figure 8 biomedicines-11-02688-f008:**
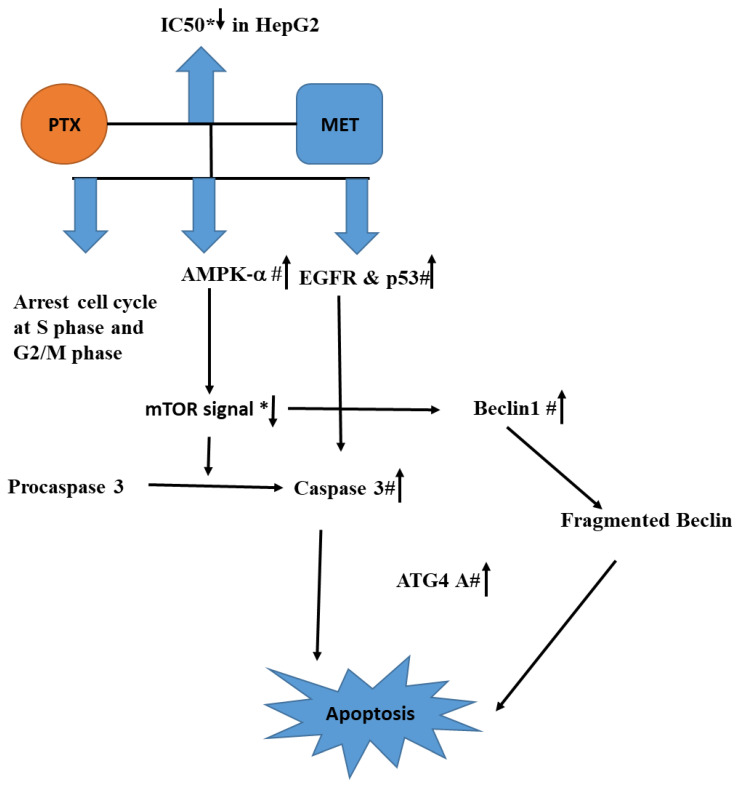
The conceptual diagram describing HepG2 treated with the combination of PXT and MET leads to the rising AMPK-α, which in turn activates caspase 3, EGFR, and p53. The combination also suppresses mTOR signaling, which stimulates the production of Beclin1 and ATG4A, resulting in cell death. Reduction or rise is indicated by an arrow with a * or a # respectively.

**Table 1 biomedicines-11-02688-t001:** Forward and reverse primer sequences for candidate genes.

Gene	Forward Primer (*5*^\^------*3*^\^)	Reverse Primer (*5*^\^------*3*^\^)
*AMPKα*	CCGAGAAGCAGAAACACGACG	CCTACCACATCAAGGCTCCGA
*EGFR*	TATTGATCGGGAGAGCCGGA	TGCGTGAGCTTGTTACTCGT
*TP53*	GCCCTCCTCAGCATCTTAT	GGTACAGTCAGAGCCAACCT
*Caspase3*	AGGTATCCATGGAGAACACTGA	GAGTCCATTGATTCGCTTCCA
*Beclin1*	AATGGTGGCTTTCCTGGACT	TGATGGAATAGGAGCCGCCA
*ATG4A*	ACTGGAGCTGGGAGAAACAA	GTCCAAACCATTCTCCAATTGAT
*GAPDH*	GACAGTCAGCCGCATCTTCT	GCGCCCAATACGACCAAATC

**Table 2 biomedicines-11-02688-t002:** IC_50_ of PXT, MET, and PXT plus 500 µg/mL MET in HepG2, HCT116, and MCF-7 human cell lines after 48 h.

	Drugs	MET	PXT	PXT and MET
Cells				
HepG2				
Range		328.9–451.4	2.249–2.885	1.132–1.782
Mean ± SD		385 ± 61.31	2.55 ± 0.318	1.42 ± 0.326
HCT116				
Range		224.6–390.1	3.403–4.164	2.624–3.292
Mean ± SD		296 ± 83.01	3.77 ± 0.381	2.94 ± 0.334
MCF7				
Range		227.2–342.1	8.267–10.88	6.302–7.545
Mean ± SD		279 ± 57.55	9.48 ± 1.308	6.89 ± 0.622

**Table 3 biomedicines-11-02688-t003:** Results of the docking of metformin (M) and paclitaxel (P) drugs on the crystal structure of targeted protein kinase.

Target Protein Name	AMPK (PDB: 4EAG)		EGFRK (PDB: 1M17)		FKBP12-Rapamycin-mTOR-FRB (PDB: 1FAP)	
Drug Name	M	P	M	P	M	P
Estimated free energy of binding (kcal/mol)	−2.76	−11.01	−4.92	−10.59	−4.76	−15.63
Estimated inhibition constant, Ki	9.53 mM	8.49 nM	248.54 μM	17.22 nM	325.8 μM	3.4 pM
vdW + Hbond + desolving energy	−3.42	−13.12	−3.75	−10.38	−3.50	−15.71
Electrostatic energy (kcal/mol)	+0.66	−0.09	−1.17	+0.58	−1.25	+0.09
Total intermolecular. energy (kcal/mol)	−2.76	−13.21	−4.92	−9.79	−4.76	−15.62
Frequency	100%	4%	97%	7%	99%	10%
Interaction surface	433.077	1387.968	330.443	1062.054	414.376	1314.83
**Target Protein Name**	**Ppiase Domain of FKBP51-Rapamycin-mTOR-FRB** **(PDB: 4DRH)**		**Beclin1(PDB: 3Q8T)**		**ATG16L1d (PDB: 5NUV)**	
**Drug Name**	**M**	**P**	**M**	**P**	**M**	**P**
Estimated free energy of binding (kcal/mol)	−4.99	−14.15	−7.41	−6.92	−4.31	−7.33
Estimated inhibition constant, Ki	221.4 μM	42.3 pM	3.67 μM	8.45 μM	689.0 μM	4.23 μM
vdW + Hbond + desolving energy	−3.84	−16.94	−3.03	−5.54	−2.53	−8.18
Electrostatic energy (kcal/mol)	−1.15	+0.09	−4.39	+0.46	−1.78	−0.18
Total intermolecular energy (kcal/mol)	−4.99	−16.85	−7.41	−5.08	−4.31	−8.35
Frequency	37%	17%	98%	1%	86%	1%
Interaction surface	438.319	1496.8	362.6	640.21	337.058	804.10
**Target Protein Name**	**ATG4A(PDB: 2P82)**		**TP53(PDB: 3DCY)**		**Caspase3 (PDB: 3GJQ)**	
**Drug Name**	**M**	**P**	**M**	**P**	**M**	**P**
Estimated free energy of binding (kcal/mol)	−5.27	−8.85	−3.25	−6.04	−3.78	−8.42
Estimated inhibition constant, Ki	136.6 μM	325.5 nM	4.13 mM	37.32 μM	1.69 mM	668.36 nM
vdW + Hbond + desolving energy	−4.29	−8.05	−2.72	−6.49	−3.50	−5.25
Electrostatic energy (kcal/mol)	−0.98	−0.15	−0.53	−0.05	−0.28	−0.82
Total intermolecular energy (kcal/mol)	−5.27	−8.20	−3.25	−6.54	−3.78	−6.07
Frequency	80%	5%	11%	1%	31%	1%
Interaction surface	354.99	827.38	332.852	773.58	293.164	723.251

**Table 4 biomedicines-11-02688-t004:** The interactions constructed between metformin (M) and paclitaxel (P) drugs and targeted protein kinase.

AMPK (PDB: 4EAG)					
M	−2.76	3	LYS126, MET84, ARG117, GLN122	3	THR86, ARG117, GLN122, ASP89
P	−11.01	4	ASP89(2), GLN122, ARG223	10	THR88, TYR120, GLN122, LYS126, VAL129(2), ILE149(2), HIS150, ARG223
EGFRK(PDB: 1M17)					
M	−4.92	4	CYS773(2), PHE771,TYR777	1	ASP776
P	−10.59	1	ASP776	10	LEU694, PHE699(2), VAL(3), ALA719, ARG817, LEU820
FKBP12-rapamycinmTOR-FRB (PDB: 1FAP)					
M	−4.76	5	LYS47(2), GLU54, TYR210(2)		GLU54(2)
P	−15.63	2	TYR82, GLU54	15	TYR26, PHE46, VAL55, ILE56, TRP59, TYR82, ILE90, PHE99, PHE2039(2), THR2098, TYR2105, HIS8, SER203, ASP37
Ppiase domain of FKBP51-rapamycin-mTOR-FRB (PDB: 4DRH)					
M	−4.99	5	GLY117, LEU119, PHE203, SER118, TYR203	3	ARG204, ALA116, **TYR2038**
P	−14.15	3	TYR57, GLN85, SER2035	17	PHE67, PHE77(3), GLN85, VAL86, LYS121, ILE122, GLU2032, ARG2036, PHE2039(2), TRP2101, TYR2105, **TRP90(2)**, ARG2036
Beclin 1, (PDB: 3Q8T)					
M	−7.41	3	GLU224, ASPB221, ALA217	3	LEU220, ASPA221(2)
P	−6.92	4	GLU216A, GLU216B, ARGA219(2)	9	LEU220, ALA215, LYS212, GLU216A (2), GLU216(B), GLU 223, ARG219A, ARG219B
ATG16L1d (PDB ID: 5NUV)					
M	−4.31	2	CYS356, GLU357	4	**PHE318**, **PHE358**, **TRP349**,CYS316
P	−7.33	2	ARG345, LYS347	7	PHE318(3), ASP319, TRP349(2), PHE358, GLU357, **PHE358**
ATG4A (PDB: 2P82)					
M	−5.27	6	ASP174(2), SER172,ASN175(2), PRO228	1	ASP174
P	−8.85	1	ASN175	4	PRO228,ARG230,ILE233(2)
TP53 (PDB: 3DCY)					
M	−3.25	4	PRO30, LEU31(2), GLU29	4	ARG15, PHE36, SER32, GLU33
P	−6.04	0	0	10	ASP28,85, LEU59, LYS63, GLN64, HIS67, TYR83, **TYR83**, **LYS63**, LYS63
Caspase3 (PDB: 3GJQ)					
M	−3.78	4	SER63, SER65(3), THR62, HIS121	1	ARG64
P	−8.42	3	THR62(2), SER65	6	ARG64, CYS163, HIS121, SER65, ASP70, GLN161

Salt bridge in red color, polar in blue color, other in orange, π–Stacking and Cation–π interactions in **bold** black color.

**Table 5 biomedicines-11-02688-t005:** Relative gene expression to control of *AMPK-α, EGFR, TP53, Caspase 3, Beclin1,* and *ATG4A* in HepG2 treated with MET, PXT, or PXT coupled with MET.

Genes	MET	PXT	PXT and MET
*AMPK-α*	5.57 ± 0.436	0.6 ± 0.073	2.92 ± 0.278
*EGFR*	0.936 ± 0.618	3.82 ± 0.583	8.44 ± 0.122
*TP53*	5.41 ± 2.06	4.568 ± 1.62	7.99 ± 2.13
*Caspase3*	3.6 ± 0.61	66.5 ± 4.14	33.0 ± 1.61
*Beclin1*	10.7 ± 1.33	4.26 ± 0.41	12.1 ± 0.431
*ATG4A*	11.06 ± 1.90	1.77 ± 0.38	11.98 ± 1.18

## Data Availability

Not applicable.

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
