# Peer review of "Regulation of Protein-Induced Apoptosis and Autophagy in Human Hepatocytes Treated with Metformin and Paclitaxel In Silico and In Vitro"

_biomedicines, 2023, doi:10.3390/biomedicines11102688_

Round 1
Reviewer 1 Report
This manuscript presents that metformin combined with paclitaxel decreased cell viability in three cancer cell lines (HepG2, HCT116 and MCF-7), increased apoptosis and cell cycle arrest in HepG2, and the effect of the combination drugs performed better than single use. The key proteins related to apoptosis and autophagy were applied to virtual docking with metformin and paclitaxel, respectively. Some of the proteins showing good interacting properties with the metformin and paclitaxel were tested in HepG2, in which real-time qPCR was used to detect the effects of the two drugs, respectively or simultaneously, on the mRNA expressions of the indicated proteins. In general, this research is of some value to clinical medication, and the methods were reasonable. However the current version of this manuscript contains lots of errors which makes it both difficult to read and scientifically inaccurate in some instances.
1.Overall it feels like the manuscript could benefit from a thorough language and scientific-term check. Once the manuscript has been tidied up and the errors have been fully corrected it would be worth resubmitting for review.
2. The images are too pixelated and stretched abnormally.
3. Three independent experiments should be conducted and data should be expressed as mean ± S.D. of values.
A sample of the many errors and suggestions are listed below:
-"In silico" and "in vitro" should be in italic in the whole manuscript, not only in title.
-cell viability were conducted in three cancer cell lines (HepG2, HCT116 and MCF-7), but the subsequent tests only in HepG2, the authors should provide some explaination for the inconsistency.
-For scientific conventions, it is better to capitalize the L in the unit name "mL" or "μL".
-in "Materials and Methods", "DMEM medium" was written as "DEMEM medium".
-The primer sequences used in the Real time qPCR for MAPKα, EGFR, Beclin1, ATG4A and GAPDH contained base mismatches.
-The statistical methods of the experimental data and descriptions should be described.
Grammers, descriptions and scientific-terms needs to be thoroughly checked.
Author Response
Dear Dr
We are grateful for the reviewers’ precise, detailed, and constructive comments that have contributed substantially to improving the presentation of our study, as well as the overall quality of the manuscript. In the following, we offer point-by-point replies to the issues that the reviewers raised regarding the original manuscript.
Please let us know as soon as possible if further clarification is needed. We look forward to learning of your decision.
#1# The manuscript could benefit from a thorough language and scientific-term check.
The manuscript's language and scientific terms have been revised and corrected.
#2# The images are too pixelated and stretched abnormally.
All the images in the manuscript have been updated to have high resolutions.
#3# Three independent experiments should be conducted, and data should be expressed as mean ± S.D. of values.
All data were carried out three experiments. So, mean ± S.D. of values, have been added to the methods used.
#4# In silico" and "in vitro" should be in italic in the whole manuscript.
The words "in silico" and "in vitro" have been italicized throughout the manuscript.
#5# Cell viability were conducted in three cancer cell lines (HepG2, HCT116 and MCF-7), but the subsequent tests only in HepG2, the authors should provide some explanation for the inconsistency.
The clarification can be found between lines 236-244 within the results section, under heading subtitle “Anti-tumor effects of paclitaxel and metformin and their association on HepG 2, HCTT116, and MCF-7 cells”.
“The IC50 values of PXT-treated HepG2 cells were 1.5 times lower than the IC50 values of PXT-treated HCT116 cells and 3.7 times lower than the IC50 values of PXT-treated MCF-7 cells. Additionally, the IC50 values of combined PXT and MET treatment were 2.1 times lower than the IC50 values of combined PXT and MET treatment on HCT116 cells, and 4.8 times lower than the IC50 values of combined PXT and MET treatment on MCF-7 cells. The most significant effects of PXT and combined PXT and MET treatments were observed in HepG2 cells compared to HCT116 and MCF-7 cells. As a result, experiments that assessed the apoptosis, cell cycle, and gene expression of autophagic and apoptotic proteins were performed on HepG2 cells exclusively”.
#6# For scientific conventions, it is better to capitalize the L in the unit name "mL" or "μL".
The unit’s name has been changed.
#7# in "Materials and Methods", "DMEM medium" was written as "DEMEM medium".
DMEM medium has been corrected.
#8# The primer sequences used in the Real time qPCR for MAPKα, EGFR, Beclin1, ATG4A and GAPDH contained base mismatches.
The primers sequences of MAPKα, EGFR, Beclin1, ATG4A and GAPDH have been revised and corrected in Table (1)
#9# The statistical methods of the experimental data and descriptions should be described.
In the last part of the material and methods, statistical methods have been incorporated.
“The viability of treated cells is given as a mean ± standard deviation (SD). By multiplying the absorbance of treated cells by 100 and dividing it by the absorbance of untreated cells, the percentage of viability was estimated. GraphPad Prism Software (version 9.0, San Diego, CA, USA) was used to calculate mean and SD, and compute the drug IC50. The flow cytometry software from Applied Biosystems automatically calculated the proportion of cells in each phase of the cell cycle, as well as the percentage of cells of necrotic and apoptotic cells”.
Reviewer 2 Report
The authors have investigated the effects of metformin on hepatocytes in combination with paclitaxel. They have also evaluated the interaction between these drugs and several proteins involved in apoptosis or autophagy, such as MAPK, EGFRK, ATG4, TP53 in silico.
Major points.
1. In Figure 4, comparison between MET vs PXT&MET and PXT vs PXT&MET should be indicated to show the additive effect of metformin.
2. In addition to gene expression (Table 5), information of protein expression will strengthen the data. The authors described the binding of metformin or paclitaxel and the proteins, however, the binding itself would not mean the upregulation of the genes.
3. The conceptual diagram describing their results would be helpful.
4. The data of liver cancer cell line other than HepG2 would be needed.
Minor points.
1. The resolution of Figures 1, 2, and 3 is low.
2. Figure 3 needs more precise explanations (Which graph is (A) or (B), The percentage of each fraction in the histogram).
The first part of 'Discussion' should be corrected.
'In the current investigation....respectively' (line 525-527)
Author Response
Dear Dr
We are grateful for the reviewers’ precise, detailed, and constructive comments that have contributed substantially to improving the presentation of our study, as well as the overall quality of the manuscript. In the following, we offer point-by-point replies to the issues that the reviewers raised regarding the original manuscript.
Please let us know as soon as possible if further clarification is needed. We look forward to learning of your decision.
#1# In Figure 4, comparison between MET vs PXT&MET and PXT vs PXT&MET should be indicated to show the additive effect of metformin.
The text and figure 4 have demonstrated significant differences of different phases of cell cycle of HepG2 treated with combination PXT & MET versus MET as well as PXT versus PXT&MET
“In the text “When HepG2 cells were given a combination of PXT and MET at an IC50 concentration, with PXT at a level of 10 μg/mL and MET at a level of 500 μg/mL, there was a noteworthy decrease of 32% in the G0/G1 phase and a significant increase of 26% in the S phase and 34% in the G2/M phase. In comparison, HepG2 cells treated with only MET and PXT were arrested in the G0/G1 phase (56% and 64%, respectively), S phase (13% and 9%, respectively), and G2/M phase (13% and 25%, respectively) (Figures 3 and 4)”.
In Figure 4:. Percentage of cells in all cell cycle phases in HepG2 treated with MET, PXT, and PXT combined with MET. PXT induces significant G2/M phase arrest and decreased percentage of cells in the G0/G1 phase, and MET slightly induces the accumulation of apoptotic cells in the subG0 phase of the cell cycle. * significant differences versus PXT, #significant differences versus MET
#2# In addition to gene expression (Table 5), information of protein expression will strengthen the data. The authors described the binding of metformin or paclitaxel and the proteins, however, the binding itself would not mean the upregulation of the genes.
It is preferable to execute protein expression by western blot, however, we did not do so because the necessary equipment was not available in our lab. However, we used molecular docking to binds proteins with PXT and MET. This will be considered in the future.
#3# The conceptual diagram describing their results would be helpful.
The diagram has been added to the end of the results section.
Figure 7: The conceptual diagram describing HepG2 treated with the combination PXT and MET leads to the rising AMPK, which in turn activates caspases3, EGFR, and p53. The combination also suppresses mTOR signaling, which stimulates the production of Beclin1 and TGA 4 1, resulting in cell death.
#4# The data of liver cancer cell line other than HepG2 would be needed.
The clarification can be found between lines 236-244 within the results section, under heading subtitle “Anti-tumor effects of paclitaxel and metformin and their association on HepG 2, HCTT116, and MCF-7 cells”.
“The IC50 values of PXT-treated HepG2 cells were 1.5 times lower than the IC50 values of PXT-treated HCT116 cells and 3.7 times lower than the IC50 values of PXT-treated MCF-7 cells. Additionally, the IC50 values of combined PXT and MET treatment were 2.1 times lower than the IC50 values of combined PXT and MET treatment on HCT116 cells, and 4.8 times lower than the IC50 values of combined PXT and MET treatment on MCF-7 cells. The most significant effects of PXT and combined PXT and MET treatments were observed in HepG2 cells compared to HCT116 and MCF-7 cells. As a result, experiments that assessed the apoptosis, cell cycle, and gene expression of autophagic, and apoptotic proteins were performed on HepG2 cells exclusively”.
#5# Resolution of Figures 1, 2, and 3 is low.
Corrections for the resolutions of three figures have been made.
#6# Figure 3 needs more precise explanations (Which graph is (A) or (B), The percentage of each fraction in the histogram).
Figure 3 has been updated to display the of A, B, C, and D. Additionally, the percentage for each fraction has been included.
Round 2
Reviewer 1 Report
The writer has implemented several effective revisions to this article,however, there remain numerous issues:
Figure 2, the calculation of the apoptosis ratio is erroneous, and the cumulative cell count within the four quadrants should approach 100%. Adding the corresponding histogram illustrating the apoptosis ratio will enhance the comprehensibility of the outcomes. Employing statistical techniques to scrutinize the data discrepancies and annotating them on a bar chart is advisable.
Figure 4, it remains unclear which statistical approach the author employs to ascertain the significance of the results.
Figure 6, it is essential to normalize and display the relative gene expression within the control group on the graph. Additionally, the AMPK expression within the pxt group should not display as negative.
In reference 44, it was indeed acknowledged that ATG4 is intricately linked to autophagy; however, solely ATG4D is correlated with caspase-3 activation. Simultaneously, the article "The insight of ATG4A in macroautophagy" highlights that ATG4A within the ATG4 family yields more limited effects in cellular autophagy compared to ATG4B, ATG4C, and ATG4D. In light of this, the rationale behind selecting ATG4A as the focus of research warrants clarification.
The author should take a comprehensive review of the manuscript and make thorough revisions, rather than addressing only the specific points I've raised. Please note that the concerns I've highlighted are just a part of the problems.
The author should take a though review of the entire manuscript to make sure the expression is accurate and scitific.
Author Response
Second Response Letter
Dear Editor:
Manuscript: biomedicines-2518894
Title of Paper: “Regulation of proteins induced apoptosis and autophagy in human hepatocytes treated with metformin and paclitaxel: In silico and in vitro”
Thank you for considering our article for publication in Biomedicines. We are grateful for the reviewers’ precise, detailed, and constructive comments that have contributed substantially to improving the presentation of our study, as well as the overall quality of the manuscript. In the following, we offer point-by-point replies to the issues that the reviewer raised regarding the original manuscript.
Please let us know as soon as possible if further clarification is needed. We look forward to learning of your decision.
Sincerely Yours,
Prof. Ehab M.M. Ali
(On behalf of all the authors)
Responses to the reviewer’s comments:
#1# Figure 2, the calculation of the apoptosis ratio is erroneous, and the cumulative cell count within the four quadrants should approach 100%. Adding the corresponding histogram illustrating the apoptosis ratio will enhance the comprehensibility of the outcomes. Employing statistical techniques to scrutinize the data discrepancies and annotating them on a bar chart is advisable.
The calculation of the apoptosis ratio has been revised and corrected. Histogram illustrated the percentage of early apoptosis has been added as Figure 3 page 9.
#2# Figure 4, it remains unclear which statistical approach the author employs to ascertain the significance of the results.
Figure 4 (now Figure 5) clarifies the comparison of the percentage of distinct phases of both treatments combined in HepG2 to each drug alone. The content on page 9 lines 305-308 has been changed.
“The percentage of cells in phase G1/G0 of HepG2 treated with combination PXT&MET decreased significantly when compared to HepG2 treated with PXT or MET alone, however the percentage of cells in S phase and G2/M phases increased in HepG2 treated with combined PXT&MET (Figures 4&5)”.
#3# Figure 6, it is essential to normalize and display the relative gene expression within the control group on the graph. Additionally, the AMPK expression within the PXT group should not display as negative.
All gene expressions have been displayed relative to the control, AMPK expression within the PXT group have been changed relative to control gene expression (Table 5 and Figure 7).
#4# In reference 44, it was indeed acknowledged that ATG4 is intricately linked to autophagy; however, solely ATG4D is correlated with caspase-3 activation. Simultaneously, the article "The insight of ATG4A in macroautophagy" highlights that ATG4A within the ATG4 family yields more limited effects in cellular autophagy compared to ATG4B, ATG4C, and ATG4D. In light of this, the rationale behind selecting ATG4A as the focus of research warrants clarification.
The significance of ATG4A as a research topic is explained on page 24 lines 658-663.
“ATG4A, a cysteine protease, can interact with GABARAP, the g aminobutyric acid receptor associated protein. GABARAP possesses a positively charged N-terminus and could bind to light chain-3 (LC3). ATG4A cleaves proLC3 to create LC3, exposing a glycine residue that permits it to bind to phosphatidyl-ethanolamine (PE) and become a component of autophagosomal membranes. ATG4A is involved in autophagy-related processes including autophagosome formation and autophagic flux control”.
The paragraph (lines 672-677 on page 24, 25) has been updated to reflect that caspase-3 cleaves ATG4D.
“The cleavage ATG4D by caspase-3 led to stimulate autophagy. When fragmented ATG4D was overexpressed in cells, it induced a change in the membrane potential of the mitochondria, leading in the release of cytochrome c and the activation of intrinsic apoptotic pathways [44]. More research is needed to determine the precise molecular pathways by which caspase-3 fragments ATG4A, resulting in the activation of both apoptosis and autophagy”.
#5# The author should take a comprehensive review of the manuscript and make thorough revisions, rather than addressing only the specific points I've raised. Please note that the concerns I've highlighted are just a part of the problems.
The manuscript has been thoroughly reviewed, and several statements have been changed throughout.
Round 3
Reviewer 1 Report
After examining the raw data, I have found several issues in apoptosis data that need to be addressed:
1, There is a lack of independently replicated experimental data in the apoptosis dataset (I guess this is only a portion of the data). Additionally, the total number of cells (only 2,000 cells per sample) analyzed is insufficient. Generally, apoptosis detection requires a minimum of 10,000 cells to be analyzed in order to obtain reliable results.
2, we noticed a significant problem with the samples or device settings, configurations... Take the control as an example, in the gate selection, fluorescence optical crosstalk was obviously observed, but it was not taken into account or addressed. This has a substantial impact on the experimental results.
3, the apoptosis data showed two distinct groups of cells within the selected analysis region. This discrepancy could be attributed to improper equipment calibration or inappropriate wavelength settings. As a result, this abnormal data lacks scientific rigor and cannot be considered reliable evidence.
In terms of the RT-qPCR mRNA expression calculations, the authers need to strictly adhere to the Livak and Schmittgen method mentioned in the manuscript. Firstly, the delta Ct value is obtained by subtracting the Ct value of the reference gene from the target gene. Then, delta-delta Ct value is obtained by the delta Ct value of the experimental group subtracted from the delta Ct value of the control group. The relative gene expression level is represented as 2^delta-delta Ct. Statistical analyses, such as t-tests and variance analysis, should be applied to the triplicate experimental data. Therefore, it is necessary to reanalyze the data accordingly.
The language and descriptions are acceptable.
Author Response
Third Response Letter
Dear Editor:
Manuscript: biomedicines-2518894
Title of Paper: “Regulation of proteins induced apoptosis and autophagy in human hepatocytes treated with metformin and paclitaxel: In silico and in vitro”
Thank you for considering our article for publication in Biomedicines. We are grateful for the reviewers’ precise, detailed, and constructive comments that have contributed substantially to improving the presentation of our study, as well as the overall quality of the manuscript. In the following, we offer point-by-point replies to the issues that the reviewer raised regarding the original manuscript.
Please let us know as soon as possible if further clarification is needed. We look forward to learning of your decision.
Sincerely Yours,
Prof. Ehab M.M. Ali
(On behalf of all the authors)
Responses to the reviewer’s comments:
After examining the raw data, I have found several issues in apoptosis data that need to be addressed:
1, There is a lack of independently replicated experimental data in the apoptosis dataset (I guess this is only a portion of the data). Additionally, the total number of cells (only 2,000 cells per sample) analyzed is insufficient. Generally, apoptosis detection requires a minimum of 10,000 cells to be analyzed in order to obtain reliable results.
Due to the lack of replicated tests, we regarded it as a preliminary study. It was indicated in the text. Page 8 lines 263-266.
Other biomarkers, including p53 and caspase 3, were used to assess apoptotic signals. Furthermore, the cell cycle results showed that treating the HepG2 cell line with this combination (PXT coupled MET) resulted in a decrease in the G0/G1 phase and an increase in the S and G2/M phases, resulting in the early onset of apoptosis.
“The preliminary study of detection of apoptosis by using annexin V-PE /7-AAD dye indicated that the early apoptotic activity of HepG2 cell lines treated with PXT combined MET was found to be 2.2 or 4 times higher than that of cells treated with paclitaxel alone or metformin alone.”
A control experiment was conducted on 2000 cells, and the findings were compared to untreated (control) cells. 2000 cells indicated the stained cells did not withdraw cells to instruments by needle. Annexin/7AAD investigations were conducted using the Guava-Nexin Reagent, and they were added in manuscript (page 3 line 115) and read by the Guava-instrument. This instrument's software is unique compared to other flow cytometry tools. The figure that is pictured below is from the PDF file for the Guava Nexin® Reagent (page 11).
This figure clarifies the stained cells by annexin and 7AAD not number of cells withdraw from needles according to the software of Guava-instrument. The flow cytometry device (Guava) may pick up 10,000 cells.
2, we noticed a significant problem with the samples or device settings, configurations... Take the control as an example, in the gate selection, fluorescence optical crosstalk was obviously observed, but it was not taken into account or addressed. This has a substantial impact on the experimental results.
The reviewer highlighted that the control was a clear example of the gate selection and fluorescence optical crosstalk. How can you be certain it is not an account? I believe the Guava-instrument distinguished itself from other flowcytometry systems.
3, the apoptosis data showed two distinct groups of cells within the selected analysis region. This discrepancy could be attributed to improper equipment calibration or inappropriate wavelength settings. As a result, this abnormal data lacks scientific rigor and cannot be considered reliable evidence.
The annexin/7AAD experiment suggests an initial inquiry and confirmation of the results, which we discussed in points 1 and 2. We investigated and evaluated apoptosis (using p53 and caspase 3).
In terms of the RT-qPCR mRNA expression calculations, the authers need to strictly adhere to the Livak and Schmittgen method mentioned in the manuscript. Firstly, the delta Ct value is obtained by subtracting the Ct value of the reference gene from the target gene. Then, delta-delta Ct value is obtained by the delta Ct value of the experimental group subtracted from the delta Ct value of the control group. The relative gene expression level is represented as 2^delta-delta Ct. Statistical analyses, such as t-tests and variance analysis, should be applied to the triplicate experimental data. Therefore, it is necessary to reanalyze the data accordingly.
The gene expression data were reanalyzed in excel sheets and sent using the Livak and Schmittgen method (see this file).